# DDX6 Helicase Behavior and Protein Partners in Human Adipose Tissue-Derived Stem Cells during Early Adipogenesis and Osteogenesis

**DOI:** 10.3390/ijms21072607

**Published:** 2020-04-09

**Authors:** Bruna Hilzendeger Marcon, Carmen K. Rebelatto, Axel R. Cofré, Bruno Dallagiovanna, Alejandro Correa

**Affiliations:** 1Laboratory of Basic Biology of Stem Cells, Carlos Chagas Institute, Fiocruz-Paraná, Curitiba 81350-010, Brazil; bruna.marcon@fiocruz.br (B.H.M.); bruno.dallagiovanna@fiocruz.br (B.D.); 2Core for Cell Technology of Pontifical Catholic University of Paraná—PUCPR, Curitiba 80215-901, Brazil; carmen.rebelatto@pucpr.br; 3Laboratory of Pharmacology Innovation, Federal University of Alagoas, Maceió 57072-970, Brazil; axel.cofre@cesmac.edu.br

**Keywords:** DDX6, human adipose tissue-derived stem cells, RNA granules, P-bodies

## Abstract

DDX6 helicase is an RNA-binding protein involved in different aspects of gene expression regulation. The roles played by DDX6 depend on the complexes associated with it. Here, for the first time, we characterize the protein complexes associated with DDX6 in human adipose tissue-derived stem cells (hASCs) and analyze the dynamics of this helicase under different conditions of translational activity and differentiation. The results obtained demonstrated that the DDX6 helicase is associated with proteins involved in the control of mRNA localization, translation and metabolism in hASCs. DDX6 complexes may also assemble into more complex structures, such as RNA-dependent granules, the abundance and composition of which change upon inhibited translational activity. This finding supports the supposition that DDX6 is possibly involved in the regulation of the mRNA life cycle in hASCs. Although there was no significant variation in the protein composition of these complexes during early adipogenic or osteogenic induction, there was a change in the distribution pattern of DDX6: the number of DDX6 granules per cell was reduced during adipogenesis and was enhanced during osteogenesis.

## 1. Introduction

DDX6 is a DEAD-box helicase that associates with different complexes and regulates the mRNA life cycle [1,2,3]. Previous studies have evaluated the role of DDX6 in gene expression regulation in stem cells from different sources. In embryonic stem cells (ESCs), DDX6 silencing increased the expression of transcripts that are targeted by microRNAs. This increase was not related to changes in mRNA stability but was related to the regulation of the translational rate [4]. DDX6 also regulated the translational rate of target mRNAs in human-induced pluripotent stem cells (hIPSCs). These targets included transcription factors and chromatin regulators involved in pluripotency control [5]. Additionally, in ESCs and IPSCs, DDX6 silencing led to a phenotype of resistance to differentiation induction, suggesting that this helicase is essential to exit from pluripotency [5]. The role of DDX6 in adult progenitors/stem cells has also been studied, and interestingly, its role in the control of stemness maintenance and differentiation induction seems to be context-specific [5]. In neural stem cells derived from hIPSCs [5] and obtained from mouse brains [6], DDX6 suppression led to impaired neuronal differentiation. Similarly, in intestinal stem cells, DDX6 depletion facilitated stem cell pool maintenance/expansion [5]. On the other hand, mesenchymal stem cells derived from hIPSCs [5], muscle progenitors [1,5] or epidermal progenitors [1] differentiated into chondrogenic, muscular and epidermal cell phenotypes, respectively, were stimulated by DDX6 silencing.

Interestingly, the activity of DDX6 may be related to the complexes with which it is associated. Wang and collaborators demonstrated that DDX6 and eIF4E are recruited by YBX1 to bind transcripts related to proliferation and stemness maintenance regulation and to stimulate their translation. On the other hand, DDX6 may associate with EDC3 and reduce the stability of KLF4 mRNA [1], which is a transcriptional factor essential for terminal epidermal differentiation [7]. As the association with different complexes can lead to different functions, other studies have analyzed the protein complexes associated with DDX6. By immunoprecipitation, including the TAP-tag method, DDX6 was found to associate with proteins from decapping machinery, such as DCP1A, EDC3, EDC4, Lsm1 and Pat1; translational machinery, such as eIF4E; translational repression factors, such as 4E-T, ataxin 2/2L (ATXN2 and ATXN2L) and LSM14 and other RNA-binding proteins (RBPs), such as YBX1, IGF2BP2 and FXR1, polyA-binding proteins and ribosomal proteins [1,2,8].

In the cell cytoplasm, the ribonucleoprotein complexes associated with DDX6 may assemble into granular structures, such as stress granules (SGs) [9,10] and P-bodies [11,12,13,14]. These organelles do not have membranes and are assembled by complexes that undergo liquid-liquid phase separation, which allows them to remain in a phase that is separate from the surrounding cytoplasm [15,16,17,18,19,20]. SGs are assembled when the cell is under stress, leading to a decrease in translational activity. Silenced mRNAs are stored in SGs and may be later translocated to translational machinery or targeted for degradation [21,22,23]. On the other hand, P-bodies are constitutive and may act as degradation [14] or storage foci [11,13,24,25]. Interestingly, in human HeLa [11,26,27] and induced pluripotent cells [5], DDX6 was found in P-bodies and was deemed necessary for the assembly and maintenance of these structures. The results from a proteomic analysis of P-bodies demonstrated that approximately one-half of the proteins found in these structures were known partners of DDX6 [13,19]. Recently, DiStefano and collaborators demonstrated that P-body homeostasis was important for the balance between stemness and differentiation in pluripotent stem cells. DDX6 and LSM14A suppression, which caused P-body disruption, impaired the exit from pluripotency, while the suppression of DCP1A, which is a component of P-bodies but is not essential for its maintenance, did not affect cell differentiation [5].

DDX6 behavior and complexes in different cell types (tumoral lineages, progenitors and pluripotent stem cells) has been previously characterized, but knowledge concerning stem cells isolated from adult tissues remains underdeveloped. Human adipose tissue-derived stem cells (hASCs) are putative sources for use in therapeutic medicine [28], and understanding the mechanisms involved in stemness maintenance and differentiation induction is important to their potential uses. We had previously observed that, in these cells, the early steps of adipogenic [29] and osteogenic [30] differentiation involve extensive gene expression regulation, with transcripts being regulated both by control of mRNA abundance and association with the polysome. Here, we focus on studying the distribution of DDX6 in hASCs and characterizing the protein complexes associated with DDX6 under the same conditions of stemness and after 24 h of adipogenic or osteogenic differentiation induction. We identified 142 proteins associated with DDX6, including 66 proteins that had not been previously found in DDX6 complexes in HEK293T cells [8], human-induced pluripotent stem cells (hiPSCs) [5] or epidermal progenitor cells [1]. The results from an ontology analysis revealed that the proteins associated with DDX6 in hASCs were mainly related to mRNA life cycle regulation. Moreover, DDX6 was observed in RNA-dependent P-body-like granules, the abundance of which changed under stress and after adipogenic or osteogenic induction.

## 2. Results

### 2.1. DDX6 Is Localized with RNA-Dependent Granules in the hASCs

The hASCs were isolated from adipose tissue obtained from liposuction surgery. To determine the phenotypes of the cells obtained, we analyzed their immunophenotypic profiles and the differentiation potential. The expression of antigens characteristic of mesenchymal stem cells and of antigens from other cell lineages was determined by flow cytometry (Appendix A). We also found that the cells had the potential for differentiation into at least two phenotypes: adipocytes and osteoblasts (Appendix A), confirming the identity of the hASCs.

Next, we investigated the expression and localization of DDX6 in the hASCs. With immunofluorescence analysis, we found that DDX6 is expressed in the hASCs, mainly localized in the cell cytoplasm, where it was both dispersed and concentrated in granules (Figure 1A). The results from different studies have demonstrated that DDX6 may be associated with RNA-containing granules [8,13] and that the association of RNA and protein is important to maintain the structure of these granules [3]. To determine whether the granules observed in the hASCs depended on RNA to maintain granular structure, we treated the cells with RNase A and analyzed the DDX6 distribution. We also analyzed the DDX6 distribution in the hASCs permeabilized but not treated with RNase A as a control. In the control (permeabilized hASCs), the DDX6 granules were maintained despite the damage caused by the permeabilization process. On the other hand, when RNase A was added, we did not observe DDX6 granules, suggesting that their maintenance is RNA-dependent (Figure 1B).

Two of the RNA-dependent granules that have been described as containing DDX6 are P-bodies [8,11,12,27] and SGs [9]. To initially characterize the protein composition of the DDX6-containing granules in the hASCs, we analyzed the colocalization of these structures with proteins that are characteristic of P-bodies and SGs. By confocal analysis, we observed that TIA1, characteristic of SGs [23], was primarily found in the nucleus of the hASCs, while DDX6 was found in the cytoplasm (Figure 2A, upper panel). On the other hand, when stress was induced (using sodium arsenite), TIA1 was partially translocated to the cytoplasm and accumulated in SGs (Figure 2A, lower panel). Under this stress condition, DDX6 granules partially colocalized with TIA1-containing SGs, suggesting that DDX6 distribution changes under oxidative stress conditions. Notably, there were DDX6 granules juxtaposed to TIA1 granules (Figure 2A, lower panel), showing a localization pattern previously described for P-bodies and SGs [22], and there were DDX6 granules found at a distance from TIA1 SGs.

To confirm this observation, the same analysis was performed with PABP, a polyA-binding protein also found in SGs. We observed that, under the control condition, PABP was dispersed in the cytoplasm of the hASCs, but it was not found in granular structures (Figure 2B, upper panel). After sodium arsenite treatment, the PABP appeared with SGs, showing a changed distribution that was similar to the pattern observed for TIA1. We found DDX6-containing granules colocalized with PABP granules, juxtaposed to PABP granules and at a distance from these structures (Figure 2B, lower panel). These observations suggested that DDX6 granules observed in the hASCs under nonstress conditions did not colocalize with SG proteins. However, when stress was induced, DDX6 granules partially colocalized with the TIA1 and PABP SGs.

Next, we analyzed whether DDX6 granules also contained DCP1A, a protein typically found in P-bodies [13,14,22]. DCP1A was dispersed in the cytoplasm of the hASCs and in granular structures that were partially colocalized with the DDX6 granules (Figure 2C, upper panel). Notably, there were granules that contained both proteins, and there were granules that were DDX6- or DCP1A-positive. The same pattern was observed after stress induction (Figure 2C, lower panel).

The results from the microscopy analysis suggested that, for the hASCs maintained in nonstress conditions, the DDX6 granules partially colocalized with DCP1A but not with TIA1 and PABP. When stress was induced, some of the observed SGs composed of TIA1 or PABP partially colocalized with DDX6 granules. Additionally, under stress conditions, DDX6 granules partially colocalized with DCP1A.

### 2.2. DDX6 Granules Observed in the hASCs Have P-Body-Like Behavior

Previous works demonstrated an increase in P-body formation upon translational activity reduction and mRNA release from polysomes [14,31], which may be induced by puromycin treatment. Moreover, by using an alkyne analogous of puromycin (namely, o-propargyl puromycin; OPP), it was also possible to label the defective ribosomal products (DRIPs) released by the disassembling translation polysomes [32,33]. Therefore, we analyzed whether treatment with OPP could affect DDX6 distribution in the hASCs. After 30 min of incubation with OPP, an increase in the mean number of DDX6 granules per cell was observed (Figure 3A and Appendix A), indicating that polysome disruption stimulated the assembly of these structures. On the other hand, we also observed that the TIAR protein was found mainly in the cell nucleus of the hASCs treated with OPP (Figure 3B, upper panel), suggesting that the granules assembled under this condition did not contain this characteristic SG protein.

Next, we investigated whether the granules assembled after OPP treatment were enriched with DRIPs. These nascent peptides released after the polysome disassembly may accumulate in SGs, and an imbalance in their clearing process may induce the formation of aberrant granules [32]. We observed that the released nascent peptides were found in the cytoplasm and in the cell nucleus. The DDX6 granules also contained but were not enriched with DRIPs (Figure 3C and Appendix A).

Then, we analyzed whether stress induction could affect the dynamics and composition of the granules. Notably, there was a reduction in the mean signal intensity of OPP-labeled nascent peptides after sodium arsenite treatment, a finding consistent with a reduction in the translational activity caused by stress (Figure 3B,C). Under this condition, TIAR partially migrated to the cytoplasm to form SGs, which accumulated DRIPs (Figure 3B, lower panel and Appendix A). DDX6 granules also had accumulated these defective nascent peptides (Figure 3C, lower panel and Appendix A).

The number of TIAR and DDX6 granules enriched with DRIPs (with a ratio of DRIPs signals within the granule/surrounding region > 1.5) per cell was determined. In the hASCs maintained under nonstress conditions, 13.8% (SEM = ± 1.825) of the DDX6 granules were enriched with DRIPs. After arsenite treatment, 41.99% (SEM = ± 1.779) of the DDX6 granules were enriched with DRIPs (Figure 3D). On the other hand, 66.42% (SEM = ± 2.979) of the TIAR SGs were enriched with DRIPs (Figure 3E).

These observations suggested that, under nonstress conditions, DDX6 was found in RNA-dependent granules, that assembly of DDX6 granules could be induced by OPP treatment and that they partially colocalized with DCP1A. After stress induction, these granules accumulated DRIPs and partially colocalized with SGs, showing a dynamic that was also consistent with P-bodies.

### 2.3. DDX6 Distribution Changes upon Adipogenic or Osteogenic Induction

The results obtained suggested that changes in the translational status of hASCs led to a redistribution of DDX6. We previously demonstrated that triggering of adipogenic or osteogenic processes led to changes in the translational profile of these cells [29,30,34]. Therefore, we investigated whether the induction of adipogenic or osteogenic differentiation for 24 h could influence DDX6 distribution in the hASCs.

With immunofluorescence, we observed that, after 24 h of adipogenic or osteogenic induction, the hASCs expressed DDX6, which was mainly dispersed in the cell cytoplasm and in granules (Figure 4A). In the hASCs induced to adipogenesis, the percentage of cells that contained DDX6 granules (Figure 4B) and the mean number of granules per cell were both reduced (Figure 4C,D). On the other hand, when osteogenesis was induced, the mean number of DDX6 granules per cell (Figure 4C,D) was increased. In addition, the number of granules per cell was variable in all culture conditions (not induced, induced to adipogenesis and induced to osteogenesis), showing an uneven distribution (Appendix A), and the mean number of granules per cell was higher than the median number of granules per cell (Figure 4C,D). Nevertheless, all the analysis performed demonstrated that the early steps of adipogenic or osteogenic induction involved a change in the distribution of DDX6 granules.

The change in the number of granules per cell upon differentiation induction led us to question whether the change was related to reduced or enhanced DDX6 expression. We had previously conducted total and polysome-associated RNA-Seq of hASCs and analyzed the results 24 h after inducing osteogenesis [30] or adipogenesis [29]. For the total mRNA fractions, there was no significant change in the abundance of DDX6 transcripts in hASCs induced to osteogenesis (log_2_(FC) = −0.17; FDR = 0.82) [30] or adipogenesis (log_2_(FC) = −0.52; FDR = 0.06) [29] compared to that in the noninduced cells. For polysome-associated fractions, there was also no significant change in the abundance of DDX6 transcript after 24 h of induction to osteogenesis (log_2_(FC) = −0.44; FDR = 0.38) [30] or adipogenesis (log_2_(FC) = −0.48; FDR = 0.07) [29] compared to noninduced cells. These observations suggest that the abundance of DDX6 transcripts and their association with the translational machinery is similar in all the conditions analyzed. Then, using Western blot analysis, we verified that the DDX6 protein expression was not changed after adipogenic or osteogenic induction (Figure 4E and Appendix A). These findings suggest that the modulation of DDX6-containing granules during differentiation was not related to the regulation of DDX6 protein abundance.

### 2.4. DDX6 Granules Observed in the hASCs after Adipogenic and Osteogenic Induction Exhibit P-Body-Like Behavior

The change in DDX6 distribution led us to question whether the granules found in hASCs upon adipogenic and osteogenic triggering had the same P-body-like composition as that previously observed in noninduced cells. Treatment of the hASCs with RNase A after 24 h of adipogenic or osteogenic induction led to the disruption of DDX6 granules, suggesting that the maintenance of their structure is RNA-dependent (Appendix A). Using OPP-labeling of nascent peptides, we also observed that these DDX6 granules did not accumulate DRIPs (Appendix A).

Next, we analyzed the distribution of proteins found in P-bodies (DCP1A) and SGs (TIA1 and PABP) in the hASCs induced to adipogenesis and osteogenesis by immunofluorescence. In both the adipogenesis- and osteogenesis-induced cells, TIA1 and PABP had a similar distribution as that observed in noninduced cells (Appendix A). TIA1 was mainly concentrated in the nucleus but did not colocalize with DDX6 granules (Appendix A). PABP was also dispersed in the cytoplasm but was not enriched in the DDX6 granules (Appendix A).

On the other hand, DCP1A was dispersed in the cytoplasm and concentrated in the granules in the hASCs induced to adipogenesis or osteogenesis for 24 h. DCP1A partially colocalized with the DDX6 granules. However, we also observed granules that were positive only for DDX6 or DCP1A (Appendix A). These observations suggested that the DDX6 granules found in the hASCs induced to adipogenesis or osteogenesis for 24 h may have a composition similar to that of the P-bodies observed in noninduced cells.

### 2.5. Complexes Associated with DDX6 after 24 h of Adipogenic and Osteogenic Induction Have Different Sedimentation Patterns

The change in DDX6 distribution may be related to the association of this helicase to complexes with different compositions and/or stabilities. Moreover, previous studies have demonstrated that DDX6 may be associated with polysomes [1,35,36]. To investigate the sedimentation pattern of DDX6, we submitted hASCs that were not induced, induced to adipogenesis or induced to osteogenesis to sucrose density gradient fractionation and performed Western blotting to identify the presence of DDX6 along the gradient.

In the noninduced hASCs, DDX6 was primarily found in the ribosome-free fraction (fraction 2), but it also comigrated with monosomes (fractions 4, 5 and 6) and light polysomes (fractions 8, 10 and 12) (Appendix A, first column). When adipogenesis was induced for 24 h, less DDX6 was found in the ribosome-free fraction, and more was found in the monosome and light polysome fraction. Under this condition, DDX6 was even identified in fraction 14, suggesting that DDX6 migrated with heavier polysomes (Appendix A, second column). A similar pattern shift was observed after 24 h of osteogenesis treatment, with a reduction in DDX6-signaling in the ribosome-free fraction and a more intense signaling, indicating comigration with monosomes and light polysomes compared to the localization signaling in the noninduced cells (Appendix A, third column). Interestingly, for the conditions analyzed, we did not observe DDX6 in the bottom of any of the tubes, where heavier complexes were expected to be found. The findings from this analysis demonstrate that, after 24 h of treatment to induce hASCs to adipogenesis or osteogenesis, there was a shift in the distribution pattern of DDX6 along the sucrose gradient, suggesting that DDX6 may be associated with complexes with different sedimentation coefficients related to different sizes, stability or composition.

### 2.6. DDX6 Is Associated with Regulators of the mRNA Life Cycle in the hASCs That Were Not Induced and Those Induced to Adipogenesis or Osteogenesis for 24 h

Previous studies demonstrated that the DDX6 function within the cell was related to the complexes associated with it [1]. Moreover, the different dynamics of DDX6 observed in the hASCs upon adipogenic or osteogenic induction may have been related to DDX6 association with different protein complexes. To analyze the protein composition of the complexes associated with DDX6 in the hASCs, we first established a protocol for immunoprecipitating DDX6 in this cell type. To optimize the methodology, we analyzed the results using variations in the protocol: the precleaning step was included or not, and two different methodologies were used for protein isolation. To verify the efficiency and specificity of the protocol used, we also compared the protein content of several fractions obtained during immunoprecipitation of DDX6: the input (cell extract), the flow-through (cell extract after the immunoprecipitation process) and the precleaning beads, and we also used the same protocol with an antibody for a nonrelated protein (anti-histone H3). All these samples were analyzed by mass spectrometry.

A range of 111 to 144 proteins were identified with the different protocols used for DDX6 immunoprecipitation, with DDX6 always having the highest signal intensity (Appendix A). In the input and flow-through analysis, the signal obtained for DDX6 corresponded to less than 0.05% of the signal obtained in the immunoprecipitation sample (Appendix A), showing that the protocol used was efficient for the enrichment of this protein. Known partners of DDX6, such as EDC3, ATXN2 and YBX1, were also found in the immunoprecipitation sample but not in the input or in the flow-through fraction (Appendix A). As unsaturated mass spectrometry analysis was performed, these results do not necessarily indicate that these proteins were not present in the input or in the flow-through fractions but does suggest that the protocol was efficient for the enrichment of complexes associated with this helicase. On the other hand, DDX6 was not found to be associated with the precleaning beads and was not found with immunoprecipitated histone H3 (Appendix A), showing that the methodology was also specific. To further confirm the results obtained by mass spectrometry analysis, we performed Western blot analysis of the immunoprecipitation of the DDX6 sample and confirmed the presence of DDX6, Dcp1a and RPL30 (Appendix A).

We used this protocol established for DDX6 immunoprecipitation followed by proteomic analysis to investigate the content of the complexes associated with DDX6 in the hASCs not induced, induced adipogenesis and induced osteogenesis for 24 h. A total of 143 proteins were identified by DDX6 immunoprecipitation in the three conditions analyzed (Appendix A). In all replicates (biological and technical), DDX6 was among the proteins with the highest signal intensity. Moreover, the proteins previously found to be associated with DDX6 complexes in different cell types were identified in the complexes obtained from the hASCs under all conditions, such as PABPC1, PABPC4, EDC3, ATXN2, ATXN2L, YBX1, LARP1, DCP1A, DCP1B and UPF1 (Appendix A). Myosins and cytoskeletal proteins, such as actin and tubulin, were also found to immunoprecipitate with DDX6 (Appendix A). The results from hierarchical clustering analysis showed that the samples were grouped first by technical and biological replicates, not by condition (Figure 5). The heatmap representation also revealed that most proteins were enriched to a similar extent among the samples, suggesting that the general protein composition in the complex with DDX6 was similar under the different conditions analyzed (Figure 5).

We also compared our results with the lists of proteins identified by DDX6 immunoprecipitation from other cell types: HEK293T cells [8], human-induced pluripotent stem cells (hiPSCs) [5] and epidermal progenitors [1] (Appendix A). Of the 142 proteins identified as associated with DDX6 in our study, 70, 42 and 24 have been identified by DDX6 immunoprecipitation in the studies of Ayache et al. [8] (HEK293T cells), Di Stefano et al. [5] (hiPSCs) and Wang et al. [1] (epidermal progenitors), respectively. On the other hand, 66 of the 142 proteins identified here have not been found in the previous studies (Appendix A, highlighted).

Despite the overall similarity among the conditions analyzed, we identified proteins differentially enriched in the DDX6 complexes after differentiation induction. ELAVL1 was more enriched in the noninduced samples than it was in adipogenesis-induced samples. MVP, U2AF1 and PRUNE2 were differentially enriched in the noninduced hASCs compared to their levels in the osteogenesis-induced samples. RPN2 was more enriched with DDX6 during the coimmunoprecipitation of osteogenesis-induced hASCs than it was in the noninduced and in the adipogenesis-induced cells. FAM98A was less enriched in the osteogenesis-induced cells than it was in the adipogenesis-induced cells (Appendix A).

Since the general compositions of the samples were similar, we performed interaction and ontology analyses with all 143 proteins identified in the different replicates. Using the STRING database, we found that most of the proteins that immunoprecipitated with DDX6 had interactions predicted with high confidence mainly for binding, catalysis and reaction (Figure 6A), with a calculated PPI (protein-protein interaction) enrichment *p* value < 1.0 × 10^−16^. According to the STRING parameters, this finding suggested that the proteins identified in our study had more interactions than would be expected for a random set of proteins of similar sizes drawn from the genome and that these proteins are at least partially biologically connected in a group. This reinforces the accuracy of the coimmunoprecipitation results. There was a notable presence of ribosomal proteins from both subunits, a finding consistent with the comigration of DDX6 with the monosomal and polysomal fractions observed in the sucrose gradient experiment (Appendix A). We further confirmed by Western blot the presence of RPL30, a protein from the large ribosomal subunit in the extracts obtained from the immunoprecipitation of DDX6 in the three conditions analyzed: noninduced, induced to adipogenesis and induced to osteogenesis (Appendix A). Notably, RPL30 had not been previously identified in the studies of Ayache (HEK293T cells, treated with RNase inhibitor), DiStefano (hiPSCs) and Wang (epidermal progenitors).

The results from the gene ontology analysis revealed that proteins identified through DDX6 immunoprecipitation were involved in RNA-binding and regulation of the RNA life cycle (Figure 6B,C and Appendix A). For the molecular function category, we observed that DDX6 complexes contained proteins related to the structural constituents of ribosomes and proteins that bind to nucleic acids (with terms specific for RNA), heterocyclic and organic cyclic compounds, cadherin and/or cell-adhesion molecules. Terms related to microfilament motors and structural molecule activities were also enriched (Figure 6B and Appendix A). When the biological process category was considered, we found that DDX6 complexes were enriched with proteins related to RNA catabolic processes (nuclear-transcribed mRNA catabolic process, RNA catabolic process, mRNA catabolic process and nuclear-transcribed mRNA catabolic process, and nonsense-mediated decay). Additionally, terms related to translation and localization (cotranslational protein targeting to membrane, SRP-dependent cotranslational protein targeting to membrane, protein targeting to ER, establishment of protein localization to the endoplasmic reticulum and protein localization to the endoplasmic reticulum) were enriched (Figure 6C and Appendix A).

## 3. Discussion

Different studies have demonstrated that DDX6 helicase regulates gene expression and is involved in the control of the stemness/differentiation process in embryonic [4,5], pluripotent-induced (iPSCs) [37], neural [6] and intestinal [5] stem cells, as well as in neural [5], epidermal [1] and muscular [1,5] progenitors. It has also been demonstrated that the role played by this helicase is related to the protein complexes associated with it [1]. Here, we analyzed, for the first time, the protein complexes associated with DDX6 in hASCs, both kept in stemness and after 24 h of induction for adipogenesis and osteogenesis. Furthermore, we analyzed the localization of the protein in cells subjected to different translational conditions and during the induction of the differentiation process.

Our results suggested that DDX6 granules found in the hASCs had a profile similar to that of P-bodies (Figure 7). The increase in the number of DDX6 granules not enriched with DRIPs after OPP treatment supported the hypothesis that the assembly of these complexes in hASCs may be related to translational machinery activity but was not necessarily related to DRIP accumulation. In HeLa cells, DRIPs may be directed to SGs, from which they are removed and degraded by the machinery of protein quality control [32]. When there is an imbalance in this clearance process, DRIPs may accumulate and generate aberrant granules, similar to pathological inclusions found in diseases such as amyotrophic lateral sclerosis [32]. Notably, Ganassi and collaborators showed that DRIPs only accumulated in SGs when the protein quality control machinery was disrupted. In the hASCs used in our study, we observed that the SGs and DDX6 granules assembled after sodium arsenite treatment were enriched with DRIPs. This result may be caused by differences in the stress protocols used. Ganassi and collaborators induced stress with MG132, which causes proteasome inhibition, for 3 h. Here, we used sodium arsenite for 30 min, which leads to oxidative stress. Further studies would allow to determinate whether this difference is indeed due to the stress-induction protocols or if the clearance process of the hASCs is different from that of other cell types and may be important for maintaining granule composition and dynamics, as previously suggested [32].

In a previous study, we demonstrated that hASCs undergoing adipogenesis or osteogenesis for 24 h undergo gene expression regulation, both by controlling mRNA abundance and through association with polysomes. Here, we observed that these scenarios are accompanied by a change in the distribution pattern of DDX6, even though no significant change is found in the abundance of the DDX6 protein (Figure 7). Di Stefano and collaborators previously demonstrated that the role played by DDX6 in ESCs was dependent on the homeostasis of the P-body structure. Additionally, DDX6-silencing in different adult stem/progenitors may have different effects on differentiation capacity [5]. Here, we observed an overall opposite effect in DDX6 distribution in the same cell type (hASCs), according to the phenotype induced, reinforcing the hypothesis that the DDX6 role and/or dynamics may be context-specific [5].

In accordance with previous observations showing no change in the level of phosphorylated eIF2α-(P)S51 in the hASCs after 24 and 72 h of adipogenic induction [34], the granules found after induced differentiation did not have SG characteristics, suggesting that the first 24 h of adipogenic and osteogenic treatment did not induce a stress response in the hASCs. The differences in DDX6 distribution during early adipogenesis and osteogenesis might be related to changes in mRNA cycling during these processes. The size, intensity and abundance of P-bodies may also increase in proliferating cells [38]. Moreover, an increase in the number of P-bodies in U2OS cells through the G1, S and G2 transitions was observed [39]. We previously verified that the proliferation of hASCs undergoing adipogenesis was reduced after 24 h [29] and 72 h [34] of induction. Therefore, the reduction in the number of DDX6 granules after 24 h of adipogenesis might be related to the cell cycle arrest observed under this condition [29]. In osteogenesis, there was no significant change in hASCs’ proliferative activity in the first 24 h (as determined by Ki67-labeling), but the number of cells increased during the differentiation process [30].

The immunofluorescence analysis performed focused in the overall behavior of the hASCs population, which is characterized as composed by a nonhomogeneous population [40,41]. As previously demonstrated, 24 h of induction treatment is not enough to compromise the hASCs specifically with the adipogenic [42,43] or the osteogenic [30,43] differentiations. Hence the use of specific markers to identify which cells have already triggered the differentiation process in this scenario would be difficult. In the future, it would be interesting to perform time-lapse experiments to follow DDX6 localization along the differentiation course to evaluate a closer relationship between this process change in DDX6 distribution.

Despite the change in the sedimentation coefficient for the DDX6 complexes after adipogenic or osteogenic induction, there was little change in the composition of the proteins associated with the helicase as identified by coimmunoprecipitation. We performed, for the first time, a proteomic analysis of the DDX6 complexes in hASCs. We identified 66 proteins that were not described in previous studies on HEK293T cells [8], hiPSCs [5] or epidermal progenitors [1]. Our analysis revealed that most of the 143 proteins found to be associated with DDX6 in hASCs were also associated with proteins involved in the mRNA life cycle. Among the proteins identified under the three conditions analyzed (noninduced, induced to adipogenesis and induced to osteogenesis), DCP1A, EDC3, IGF2BP2, LSM14A, LSM14B, MOV10, STAU1 and UPF1 were previously found in P-bodies [13,22,44]. On the other hand, there was a significant presence of ribosomal proteins (from both large and small subunits), a finding consistent with the comigration of DDX6 with monosomes and light polysomes, as observed in the sucrose gradient experiment. It was also noticeable that these ribosomal proteins are not generally enriched in P-bodies [13,45]. These results suggest that the protocol was efficient for isolating the proteins found in complexes with DDX6. The identification of myosins and cytoskeleton proteins in the immunoprecipitation of DDX6 [8] and in the proteomic analysis of P-bodies [13] was also previously described. The colocalization of MYO6 and P-bodies was also observed by immunofluorescence in a previous work, which might suggest that this protein may connect these granules to the cytoskeleton [13].

The results obtained suggested that the complexes associated to DDX6 in hASCs have a similar protein content in the conditions analyzed. Nevertheless, it is possible that these proteins may be differentially distributed in specific complexes. To analyze these specific complexes, it would be necessary, for example, to separate fractions of complexes or to perform the immunoprecipitation of other proteins to characterize their complexes and to compare with DDX6 complexes. Moreover, six proteins were identified as differentially enriched before and after adipogenic and osteogenic induction. U2AF1 is an RBP that is part of the splicing machinery [46]. Interestingly, the association of DDX6 with proteins of the splicing system was previously reported [8]. FAM98A has low domains of low complexity in the C-terminal region and has been found in SGs but not in P-bodies [47]. In addition, it also associates with proteins such as ATXN2, ATXN2L, DDX1 and NUFIP2 [47], which were also found to immunoprecipitate with DDX6 from the hASCs. ELAVL1 (also known as HuR) is an RBP that was previously found to be associated with DDX6 in HEK293T cells [8]. In neuroblastoma cells, PRUNE2 interacts with the AKT pathway [48]. Activation of the AKT pathway stimulates osteogenic differentiation [49,50], and interestingly, we observed a differential association of PRUNE2 with DDX6 in the hASCs induced to osteogenesis for 24 h. In cancer cells, RPN2 was described as being involved in the regulation of proliferation and migration [51,52], and in human airway smooth muscle cells, the protein MVP was found to be involved in growth/survival signaling pathways [53]. The action of MVP in these pathways was related to its association with MYH9 [53], a protein we also identified in the DDX6 complexes in the hASCs. Nevertheless, further investigation to confirm the association of DDX6 with these proteins and a possible differential association among them is necessary to investigate the possible role of these complexes.

## 4. Materials and Methods

### 4.1. Subjects and Cell Culture

hASCs were isolated from adipose tissue obtained from donors who submitted to liposuction surgery. Tissue collection and cell isolation were performed after donors or parent/legal guardians (for donors under the age of 18 years old) provided informed consent, in accordance with the guidelines for research involving human subjects and with the approval of the Ethics Committee of Fundação Oswaldo Cruz, Brazil (approval number CAAE: 48374715.8.0000.5248, 04 November 2015). The information about the sex, age, height and body mass index (BMI) of each donor is described in Table 1.

hASC isolation was accomplished as previously described [29,54]. One-hundred milliliters of adipose tissue was washed with sterile phosphate-buffered saline (PBS) and incubated with type I collagenase (0.4 mg/mL) (Gibco, Grand Island, NY, USA) at 37 °C for 30 min under constant shaking. The digested tissue was filtered through a 100-µm mesh filter (Jet Biofil, China) and centrifuged (800× *g*, 10 min, 8 °C), and the cell pellet was treated with a hemolysis buffer of pH 7.3 (0.83% ammonium chloride, 0.1% sodium bicarbonate and 0.04% EDTA) for 10 min to remove erythrocytes. The isolated cells were washed, plated at 1 × 10^5^ cells/cm^2^ with DMEM-F12 (Gibco, Grand Island, NY, USA) supplemented with 10% fetal bovine serum (FBS), penicillin (100 units/mL) and streptomycin (100 μg/mL) and incubated in a humidified incubator at 37 °C and 5% CO_2_. After 24 h, the nonadherent cells were removed, and the culture medium was changed every 2 days. Upon reached 80–90% confluence, the cells were harvested and plated as passage 1 using DMEM (Gibco, Grand Island, NY, USA) supplemented with 10% fetal bovine serum (FBS), penicillin (100 units/mL) and streptomycin (100 μg/mL). All experiments were performed with cells from passages 4–6.

The isolated hASCs were characterized by immunophenotyping and analysis of the differentiation potential, according to the minimal criteria for mesenchymal stem cell identity, as established by the International Society for Cellular Therapy [55]. The cells were harvested and incubated in blocking solution (PBS; 1% bovine serum albumin—BSA) at 4 °C for one hour. Then, the cells were incubated for one hour at 4 °C in the dark with the following antibodies: FITC-conjugated anti-CD90 (Thy1) (BioLegend, San Diego, CA, USA); anti-CD34, anti-CD31 and anti-CD19 (E-Bioscience, Carlsbad, CA, USA); APC-conjugated anti-CD73, anti-CD45 and anti-HLA-DR (E-Bioscience, Carlsbad, CA, USA) and PE-conjugated anti-CD105 (E-Bioscience, Carlsbad, CA, USA), anti-CD11b and anti-CD140b (BD, San Diego, CA, USA). The antibodies were diluted in blocking solution, and mouse IgG antibodies (FITC, APC and PE) (BD, San Diego, CA, USA) were used as negative controls. After incubation, the cells were washed once with PBS, and the data were acquired on a FACSCanto II instrument (BD, San Diego, CA, USA). For each sample, data on at least 10,000 events were collected and analyzed with FlowJo^®^ v.10 software (FlowJo, LLC) (BD, San Diego, CA, USA).

To assess the adipogenic and osteogenic differentiation potentials of the isolated hASCs, we used hASCs in passages 4–5. The differentiation treatment was initiated when the cells reached 80% confluence. The adipogenic differentiation was induced using hMSC adipogenic differentiation medium for 3 days and hMSC adipogenic maintenance medium for 4 days (hMSC adipogenic bullet kit, Lonza, Walkersville, MD, USA), and this cycle was repeated for a total of 28 days. For osteogenic differentiation, the cells were treated with an osteogenic differentiation bullet kit (Lonza, Walkersville, MD, USA) medium for 21 days. The efficiency of adipogenic differentiation was determined by assessing the cytoplasmic accumulation of triglycerides with Nile red staining. The osteogenic differentiation was assessed using the OsteoImage™ mineralization assay (Lonza, Walkersville, MD, USA), which contains a reagent that specifically binds and stains in fluorescent green the hydroxyapatite portion of the bone-like nodules deposited by the cells.

### 4.2. Western Blot Analysis

The cells were washed with PBS and scraped after the addition of a buffer solution (160 mM Tris-HCl; pH 6.8; 4% SDS; 10% b-mercaptoethanol; 24% glycerol and 0.02% bromophenol blue) that promotes cell lysis and denatures proteins. The cell extract was collected, incubated at 95 °C for 10 min and subjected to SDS-PAGE. Then, the proteins were transferred to nitrocellulose membranes. The nitrocellulose membrane was blocked (1-h incubation with 5% milk in Tris-buffered saline—TBS), washed and incubated with the primary antibodies (anti-DDX6, MBL, Japan, PD009, 1:2500; anti-DCP1A, Santa Cruz, Dallas, TX, USA, sc-100706, 1:200 and anti-RPL30, Abcam, Cambridge, UK, ab170930). After incubation with the suitable secondary antibodies (anti-mouse IgG-alkaline phosphatase, Sigma, St. Louis, MO, USA, 1:1000 and anti-rabbit IgG-peroxidase, produced in goat, Sigma, St. Louis, MO, USA, 1:2500), the membranes were analyzed with AP buffer, BCIP e NBT (Promega, Madison, WI, USA) or with a Novex^®^ ECL HRP chemiluminescent kit (Invitrogen, Carlsbad, CA, USA). The signal intensity was quantified with ImageJ software [56].

### 4.3. Immunofluorescence and DDX6 Granule Quantification

For immunofluorescence, hASCs were cultivated in Nunc™ Lab-Tek™ chamber slide systems (Thermo Fisher Scientific, Rochester, NY, USA), fixed with 4% paraformaldehyde (20 min) and washed with PBS. The cells were incubated with permeabilization solution (0.5% Triton X-100 in PBS) for 10 min, washed with PBS and incubated with blocking solution (3% bovine serum albumin—BSA) for 1 h at room temperature. The cells were incubated with a primary antibody (anti-DCP1A, Santa Cruz, Dallas, TX, USA, sc-100706, 1:50; anti-PABP, Santa Cruz, Dallas, TX, USA, sc-32318, 1:50; anti-DDX6, MBL, Japan, PD009, 1:400; anti-TIA1, produced in mouse, kindly provided by Dr. Fabíola Holetz of Instituto Carlos Chagas, 1:400 and anti-TIAR, Cell Signaling, 8509S, 1:400) diluted in blocking solution for 1 h at room temperature and then washed with PBS. The secondary antibody (anti-mouse IgG conjugated with Alexa Fluor^®^-546, Invitrogen, Carlsbad, CA, USA, A11003, 1:600; anti-mouse IgG conjugated with Alexa Fluor^®^-488, Invitrogen, Carlsbad, CA, USA, A11001, 1:800; anti-rabbit IgG conjugated with Alexa Fluor^®^-546, Invitrogen, Carlsbad, CA, USA, A11010, 1:600 or anti-rabbit IgG conjugated with Alexa Fluor^®^-488, Invitrogen, Carlsbad, CA, USA A11008, 1:800) was also diluted in blocking solution, and the incubation was also performed at room temperature for 1 h protected from light. The nuclei were stained with DAPI for 10 min. After washing, the cells were maintained in PBS. The analysis was performed using a DMI6000B microscope (Leica, Germany), and the images were captured with LAS AF software (Leica, Germany). Confocal microscopy images were acquired using a Leica SP5 AOBS confocal microscope (Leica, Germany).

To quantify the DDX6 granules, 20 images of the cells in each condition were captured. A cutoff was established using the threshold tool of the LAS AF software (Leica, Germany) based on light intensity such that only the granules (brighter structures) were exhibited. The same threshold was applied for all the conditions, and the number of granules in each cell was counted manually.

### 4.4. Assessment of the Dependence on RNA for DDX6 Granule Maintenance

The cells in culture were washed twice with PBS and incubated with permeabilization solution (0.5% Triton X-100 in PBS) for 2 min at 37 °C. The permeabilization solution was removed, and the cells were washed twice with PBS and incubated with RNase A (10 mg/mL) for 5 min at room temperature. Then, the cells were incubated with fixation solution (4% paraformaldehyde in PBS) for 20 min, washed with PBS and subjected to immunofluorescence to identify the DDX6 localization. As a control, the immunofluorescence analysis was also performed in cells subjected only to the permeabilization process.

### 4.5. Treatment with o-propargyl Puromycin and Localization of Nascent Peptides

To evaluate DDX6 distribution after translation was inhibited with puromycin and to identify nascent peptide localization, we used Click-it Plus OPP protein synthesis assay Alexa 488 (Invitrogen, Eugene, OR, USA). The cells in culture were incubated with 20-µM o-propargyl puromycin (OPP, diluted in culture medium) for 30 min (37 °C), washed once with PBS and incubated with fixation solution (4% paraformaldehyde in PBS) for 15 min. The cells were washed, submitted to permeabilization (incubation with 0.5% Triton X-100 in PBS for 10 min) and washed again with PBS. Then, we performed the reaction for fluorescence-labeling of nascent peptides containing OPP according to the manufacturer’s instructions.

After OPP-labeling, the hASCs were subjected to immunofluorescence for the identification of DDX6 and TIAR localization. The samples were analyzed by confocal microscopy (Leica SP5 AOBS, Leica, Germany).

To evaluate nascent peptide localization under stress, the same experiment was performed with (1) hASCs treated only with OPP for 30 min and (2) hASCs treated with both OPP and sodium arsenite (0.5 mM) for 30 min.

### 4.6. Quantification of DRIPs

To measure the extent of nascent peptides labeled with OPP in granules containing DDX6 and TIAR, 5 images of cells under each condition were captured by confocal microscopy (Leica SP5 AOBS, Leica, Germany). The images were analyzed using Columbus™ Image Data Storage and Analysis software (PerkinElmer, Waltham, MA, USA).

The methodology applied was based on the study of Ganassi and collaborators [32]. Briefly, the nuclei and cytoplasm were delimited based on DAPI and OPP fluorescent staining, respectively. The DDX6 and TIAR granules were identified based on fluorescence intensity using the common threshold tool. Then, the surrounding cytoplasm (which consisted of the cytoplasm region but not the granule region) was delineated. The mean OPP intensity was measured in the granules and in the cytoplasm surrounding region. OPP enrichment was calculated as the ratio of the mean intensity in each granule to the mean intensity in the surrounding region. The obtained data were plotted in frequency distribution histograms. Granule ratios of 1.5 or greater were considered to be enriched in OPP-labeled nascent peptides, as described by Ganassi and collaborators [32]. Bar graphs showing the percentages of OPP-enriched granules per cell were also plotted.

### 4.7. Sucrose Density Gradient Separation

The cells were treated with 100-µg/mL cycloheximide diluted in culture medium (Sigma, St. Louis, MO, USA) (10 min, 37 °C), harvested, washed twice (100-µg/mL cycloheximide in PBS) and centrifuged (700× *g*, 5 min). The cell pellet was incubated with lysis buffer (15-mM Tris-HCl, pH 7.4; 15-mM MgCl_2_; 300-mM NaCl; 100-µg/mL cycloheximide and 1% Triton X-100) for 10 min on ice and centrifuged (12,000× *g*, 10 min, 4 °C), and the supernatant was loaded on a sucrose density gradient (10–50%), which was prepared with a model 108 Gradient Master from BioComp (Fredericton, NB, Canada). Then, the samples were centrifuged at 150,000× *g* (SW40 rotor, HIMAC CP80 WX HITACHI, Tokyo, Japan) for 2.5 h at 4 °C. The sucrose fractions were collected using an ISCO system (ISCO Model 160 Gradient Former Foxy Jr. Fraction Collector, Teledyne Isco, St. Lincoln, NE, USA) with an associated UV detector (Teledyne Isco, St. Lincoln, NE, USA). The absorbance was recorded at 275 nm for use in determining the polysome profile.

### 4.8. Immunoprecipitation of DDX6

The protocol for immunoprecipitation was based on the methodology previously described by Jain et al. [16] but with modifications. The cells were washed with PBS, lysis buffer (50-mM Tris-HCl, pH 7.4; 100-mM potassium acetate; 2-mM magnesium acetate; 0.5-mM dithiothreitol; 1% Nonidet P-40; 65-units/mL RNase OUT RNase inhibitor—Invitrogen, Carlsbad, CA, USA, and cOmplete Mini EDTA-free protease inhibitor cocktail tablets, Roche, Mannheim, Germany) was added and the cells on ice were scraped. The cell lysate was collected, transferred to a centrifuge tube, passed through a syringe needle twice and subjected to sonication (two cycles of 10 s with 20-s intervals on ice). After centrifugation (5 min, 1000× *g*, 4 °C), the supernatant was transferred to a new tube and centrifuged (20 min, 10,000× *g*, 4 °C). The supernatant was transferred to a new tube, and protein A Dynabeads (Invitrogen, Carlsbad, CA, USA) were added to the cell extract and incubated for 3 h at 4 °C under gentle shaking for precleaning. The precleaning beads were removed, and an antibody was added (anti-DDX6, MBL, Japan, PD009) and incubated for 16–17 h at 4 °C under gentle shaking. The magnetic beads were added and incubated for 3 h at 4 °C under gentle shaking. The magnetic beads were collected and submitted to sequential washing steps: three cycles of incubation for 5 min each with washing buffer 1 (20-mM Tris-HCl, pH 8, and 200-mM sodium chloride); incubation for 5 min with washing buffer 2 (20-mM Tris-HCl, pH 8, and 500-mM sodium chloride); incubation for 2 min with washing buffer 3 (50-mM Tris-HCl, pH 7.4; 100-mM potassium acetate; 2-mM magnesium acetate; 0.5-mM dithiothreitol; 1% Nonidet P-40; 2-M urea and cOmplete Mini EDTA-free protease inhibitor cocktail tablets, Roche, Mannheim, Germany) and incubation for 5 min with washing buffer 4 (50-mM Tris-HCl, pH 7.4; 100-mM potassium acetate; 2-mM magnesium acetate; 0.5-mM dithiothreitol and cOmplete Mini EDTA-free protease inhibitor cocktail tablets, Roche, Mannheim, Germany). The samples were transferred to a new tube, and the proteins were extracted with TRIzol (Ambion, Carlsbad, CA, USA)) according to the manufacturer’s instructions.

In the initial test to establish the immunoprecipitation methodology and to verify the efficiency of the process, we performed this protocol with three variations: (1) with the precleaning step and with protein extraction conducted with denaturing buffer (160-mM Tris-HCl, pH 6.8, 4% SDS, 10% b-mercaptoethanol, 24% glycerol and 0.02% bromophenol blue); (2) without the precleaning step and with protein extraction using a denaturing buffer and (3) without the precleaning step and with a protein extraction using TRIzol (Ambion, Carlsbad, CA, USA). After this analysis, we immunoprecipitated DDX6 after a precleaning step and extraction with TRIzol (Ambion, Carlsbad, CA, USA) as described.

Immunoprecipitation was performed with samples from two donors (biological replicates) and with two technical replicates from each donor. The precleaning bead content was also analyzed and used as a negative control.

### 4.9. Proteomic Analysis

For the proteomic analysis, the proteins were subjected to SDS-PAGE, and the gel was stained with silver. The stain was removed by incubation with a discoloration buffer (50-mM sodium thiosulfate and 15-mM potassium ferricyanide) for 10 min at 25 °C and by washing with ABC buffer (50-mM ammonium bicarbonate). Then, the gel was dehydrated with ethanol, dried (using the SpeedVac system, Thermo Fisher, Rochester, NY, USA) and incubated with a reduction buffer (10-mM dithiothreitol in ABC buffer) for 1 h at 56 °C. The supernatant was discarded, and the gel was incubated with alkylation buffer (55-mM iodoacetamide in ABC buffer) for 45 min at 25 °C. The gel was washed with ABC buffer and dehydrated with ethanol twice. Trypsin (12.5-ng/µL in ABC buffer) was added to the gel and incubated for 20 min, and then, the excess trypsin was removed, ABC buffer was added and the samples were incubated in this solution for 16–18 h at 37 °C. The extraction was performed by adding TFA solution (3% trifluoroacetic acid and 30% acetonitrile) for 10 min at 25 °C, and the supernatant was collected. Then, 100% acetonitrile was added, and the supernatant was collected. The extraction process was repeated once. The volume of the supernatant collected was reduced to 10–20% (using the SpeedVac system, Thermo Fisher Scientific, Rochester, NY, USA), and the peptides obtained were purified with StageTips-C18.

The mass spectrometry analysis was performed with a Thermo Fisher LTO Orbitrap XL system (Thermo Fisher Scientific, Rochester, NY, USA) (at the Mass Spectrometry Facility, RPT02H, Fiocruz, Curitiba, PR, Brazil), and the peptides were identified by MaxQuant software (version 1.6.1.0, Germany).

The proteins identified were analyzed using Perseus (version 1.6.2.3, Germany). Proteins identified only by site, reverse and potential contaminants were removed. The proteins identified in the analysis of the precleaning beads were also removed from the coimmunoprecipitation list because of their potential for unspecific binding. We considered as identified protein in a sample when having at least 2 unique peptides in at least one of the replicates.

The intensity of the signals was used to quantify the proteins in each sample. Intensity signal values were converted to a logarithmic scale, and missing values were imputed from a normal distribution for each replicate. Unsupervised hierarchical grouping was performed using Morpheus software based on the Euclidian distance used as a dissimilarity measure. Proteins differentially associated with DDX6 in the conditions analyzed were identified by ANOVA (*p* value < 0.05), followed by a post hoc test (FDR = 0.05).

Gene ontology analyses were performed using gProfiler [57], and protein-protein interactions were mapped using STRING (version 11.0) [58]. To compose the interaction map, the following parameters were applied: connections between knots represented molecular actions, based on text mining, experiments and databases as information sources and a minimal score of 0.7 (high confidence).

### 4.10. Statistical Analysis

All experiments were performed with three biological replicates (samples from three different donors). Except for the coimmunoprecipitation analysis, which was performed with samples from two donors (biological replicates) and with two technical replicates from each donor. A sample from one replicate was subjected to the sucrose density gradient.

Statistical analysis was performed using GraphPad Prism (version 7.01, GraphPad Software, San Diego, CA, USA). The Shapiro-Wilk normality test was used. For experiments with two conditions and a non-normal distribution, an unpaired Mann-Whitney test was used. For experiments with three or more conditions and normal distribution, one-way ANOVA for multiple comparisons and Tukey’s post hoc test were used. For experiments with three or more conditions and non-normal distribution, the Kruskal-Wallis test and Dunn’s post-test for multiple comparisons were used.

## 5. Conclusions

The data provided here showed that, in hASCs, DDX6 associates with proteins that regulate the mRNA life cycle, as was previously described for other cell types. These complexes may assemble mRNA-dependent granules that have at least some features similar to those of P-bodies. The pattern of distribution of the DDX6 granules changed when the cells were subjected to treatments that affected translational activity and during early adipogenic or osteogenic differentiation. Further investigation to determine the transcripts that may be associated with DDX6 complexes is important for identifying the genes that might be affected during the processes described herein.

## Figures and Tables

**Figure 1 ijms-21-02607-f001:**
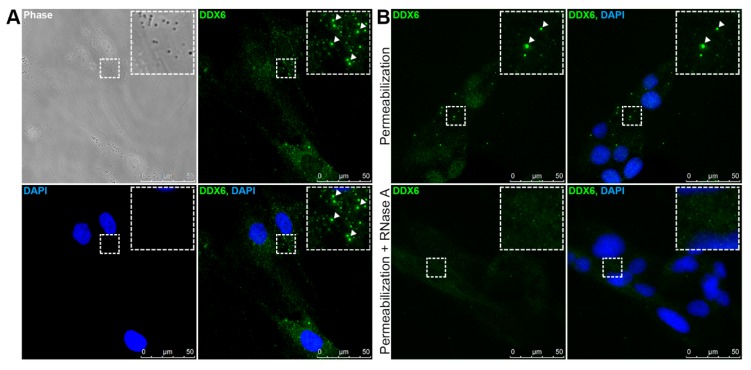
Human adipose tissue-derived stem cells (hASCs) have RNA-dependent DDX6 granules. (**A**) Results from the immunofluorescence analysis of DDX6 localization in the hASCs. The helicase is dispersed in the cell cytoplasm and accumulates in granules. (**B**) Results from the immunofluorescence analysis of DDX6 in permeabilized or permeabilized and RNase A-treated hASCs. After RNase action, the DDX6 granules were disassembled. White arrows: granules that contain DDX6. Nuclei were stained with DAPI.

**Figure 2 ijms-21-02607-f002:**
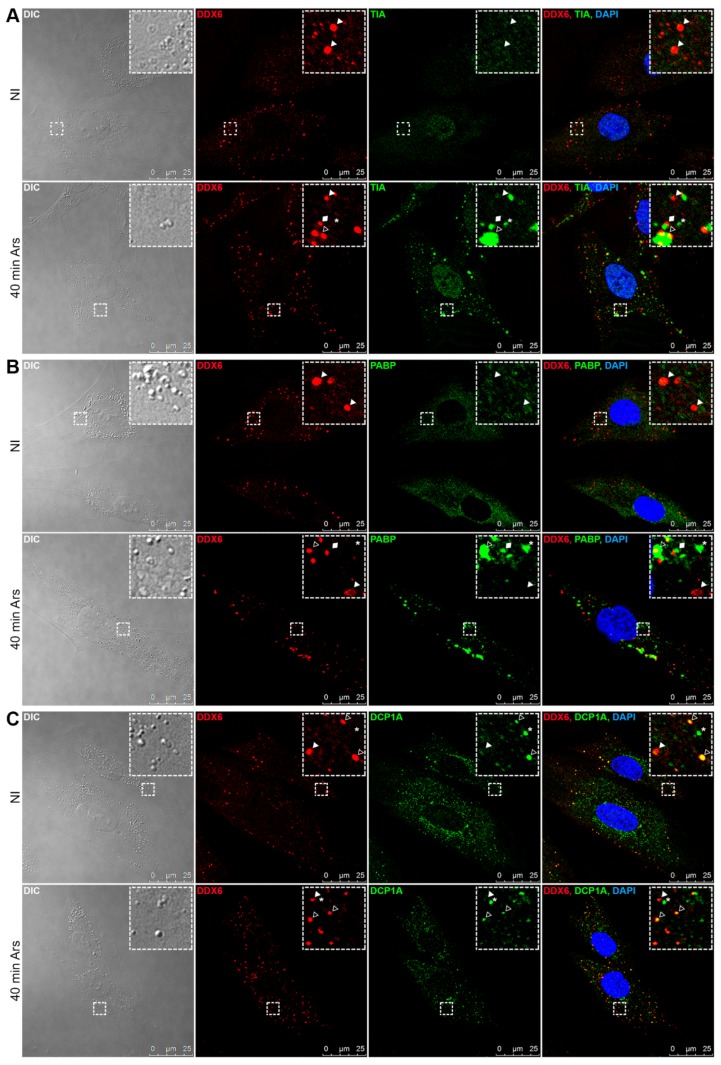
Localization of DDX6 and stress granule (SG) or P-body proteins in the hASCs not induced by stress and after stress induction. The distribution of the proteins in the hASCs maintained in the noninduction control medium (NI) and hASCs in medium treated with sodium arsenite (40 min Ars) was analyzed by immunofluorescence and confocal microscopy. (**A**) In the NI hASCs, TIA1 was found mainly in the cell nucleus. After stress induction, TIA1 migrated to the cytoplasm and aggregated in the SGs that partially colocalized with DDX6 granules. (**B**) PABP was found in the cell cytoplasm. After stress induction, PABP was also concentrated in the SGs that partially colocalized with DDX6 granules. (**C**) Under both the NI and stress conditions, DDX6 partially colocalized with DCP1A. Notably, under both conditions, granules containing only DDX6 or only DCP1A were found. White arrows: granules that contained only DDX6; asterisks: granules that contained only TIA1 (**A**), PABP (**B**) or DCP1A (**C**); hollow arrows: granules containing both DDX6 and TIA1 (**A**), PABP (**B**) or DCP1A (**C**); and rhombus: granules with juxtaposed DDX6 and TIA1 (**A**) or PABP (**B**). Nuclei were stained with DAPI.

**Figure 3 ijms-21-02607-f003:**
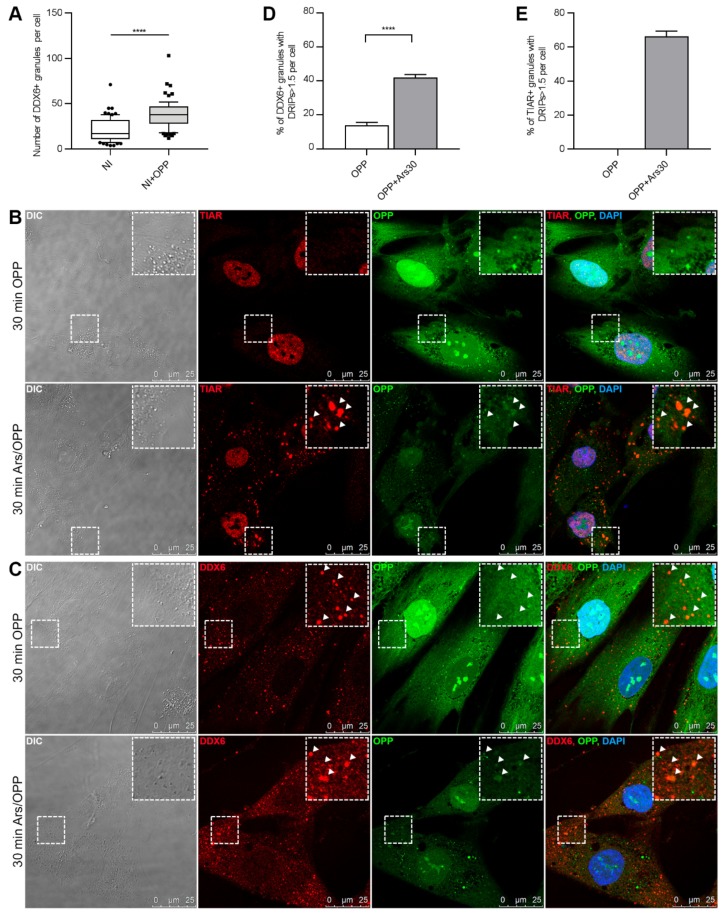
DDX6 granule assembly was regulated by translational activity. (**A**) Quantification of the mean number of DDX6 granules per cell before (NI) and after treatment of the hASCs with o-propargyl puromycin (OPP) (NI+OPP). Seventy cells per condition were analyzed (*n* = 70). Quartile 10–90%; Mann-Whitney test: **** *p* < 0.0001. (**B**) Images from immunofluorescence analysis by confocal microscopy of TIAR and DRIP localization in the hASCs treated with OPP only (30-min OPP) or OPP and sodium arsenite (30-min Ars/OPP). SGs containing TIAR showed accumulated DRIPs but only under stress conditions. White arrows: granules. Nuclei were stained with DAPI. (**C**) Results from immunofluorescence analysis by confocal microscopy of DDX6 and DRIP localization in the hASCs treated with OPP only (30 min of OPP incubation) or OPP and sodium arsenite (30 min of Ars/OPP incubation). DDX6 granules were found under both stress and nonstress conditions; however, they accumulated DRIPs only after stress induction. White arrows: granules. Nuclei were stained with DAPI. (**D**) Quantification of DDX6 granules enriched with DRIPs in the hASCs treated with only OPP (OPP) or with OPP and sodium arsenite (OPP+Ars30). The bar graph shows the percentages of DDX6 granules enriched with DRIPs (granule/surrounding region signal ratio > 1.5) per cell. At least 34 cells were analyzed per condition; standard error of the mean (SEM); Mann-Whitney test: **** *p* < 0.0001. (**E**) Quantification of TIAR granules enriched with DRIPs in the hASCs treated with only OPP (OPP) or OPP and sodium arsenite (OPP+Ars30). The cells treated with only OPP did not have assembled TIAR granules. The bar graph shows the percentages of TIAR granules enriched with DRIPs (granule/surrounding region signal ratio > 1.5) per cell. Thirty-two cells were analyzed; standard error of the mean (SEM).

**Figure 4 ijms-21-02607-f004:**
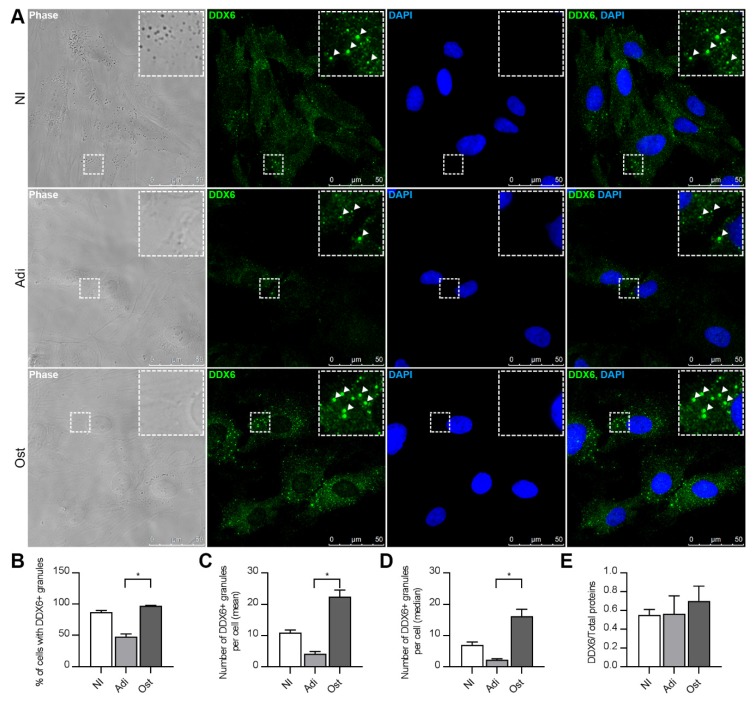
The DDX6 granule distribution changed upon adipogenic and osteogenic differentiation induction. (**A**) Results from the immunofluorescence analysis with microscopy showing DDX6 localization in the hASCs maintained in noninduction medium (NI) or induced to adipogenesis (Adi) or osteogenesis (Ost) for 24 h. Notably, the granular pattern was changed in cells stimulated to differentiate. Nuclei were stained with DAPI. (**B**) Percentages of the cells that contained DDX6 granules in hASCs maintained in noninduction medium (NI) or induced to adipogenesis (Adi) or osteogenesis (Ost) for 24 h. (**C**) Mean number of DDX6 granules per hASC maintained in noninduction medium (NI) or induced to adipogenesis (Adi) or osteogenesis (Ost) for 24 h. (**D**) Median number of DDX6 granules per hASC maintained in noninduction medium (NI) or induced to adipogenesis (Adi) or osteogenesis (Ost) for 24 h. (**E**) Quantification of DDX6 protein expression (by Western blotting) in hASCs maintained in noninduction medium (NI) or induced to adipogenesis (Adi) or osteogenesis (Ost) for 24 h. (**B**–**D**) Samples from three different donors were analyzed (*n* = 3); standard error of the mean (SEM); Kruskal-Wallis test and Dunn’s multiple comparisons post-test: * *p* < 0.05.

**Figure 5 ijms-21-02607-f005:**
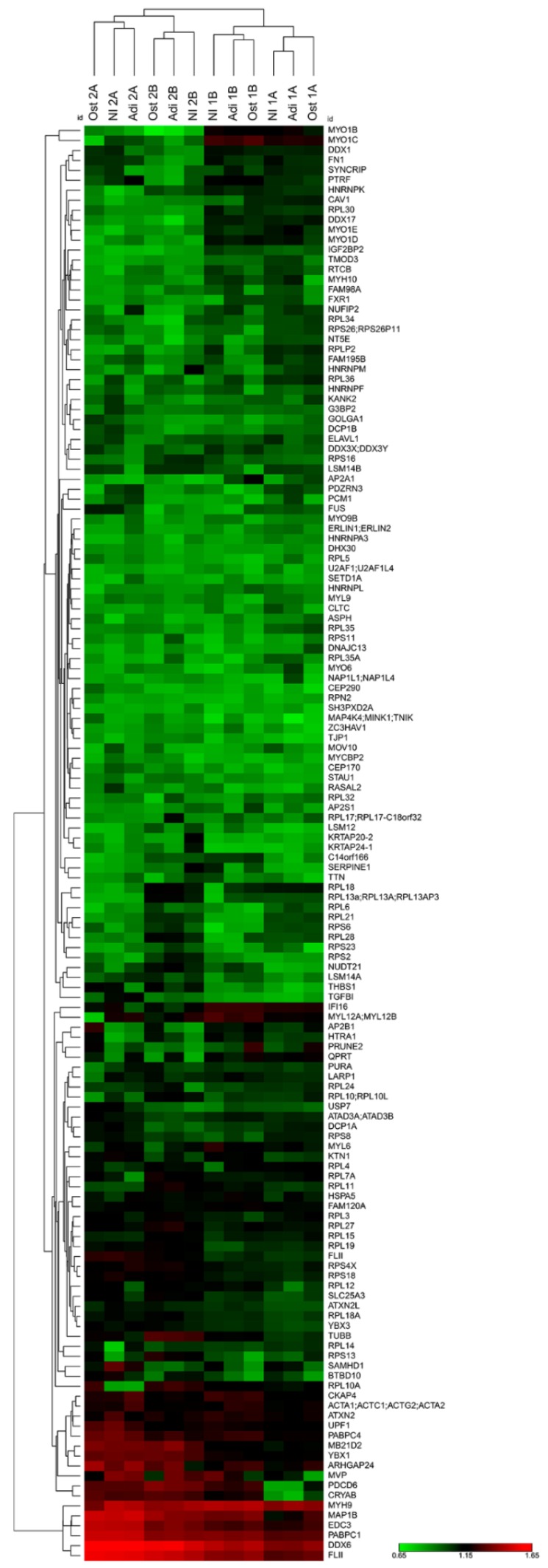
Heatmap and hierarchical clustering of the proteins identified by coimmunoprecipitation with DDX6 in the hASCs not induced or induced to adipogenesis or osteogenesis for 24 h. Samples were clustered first by technical (A and B) and then by biological (donors 1 and 2) replicates, not by condition (NI, Adi and Ost). The heatmap shows that the general composition of DDX6-associated complexes is similar during the first 24 h of adipogenic or osteogenic induction.

**Figure 6 ijms-21-02607-f006:**
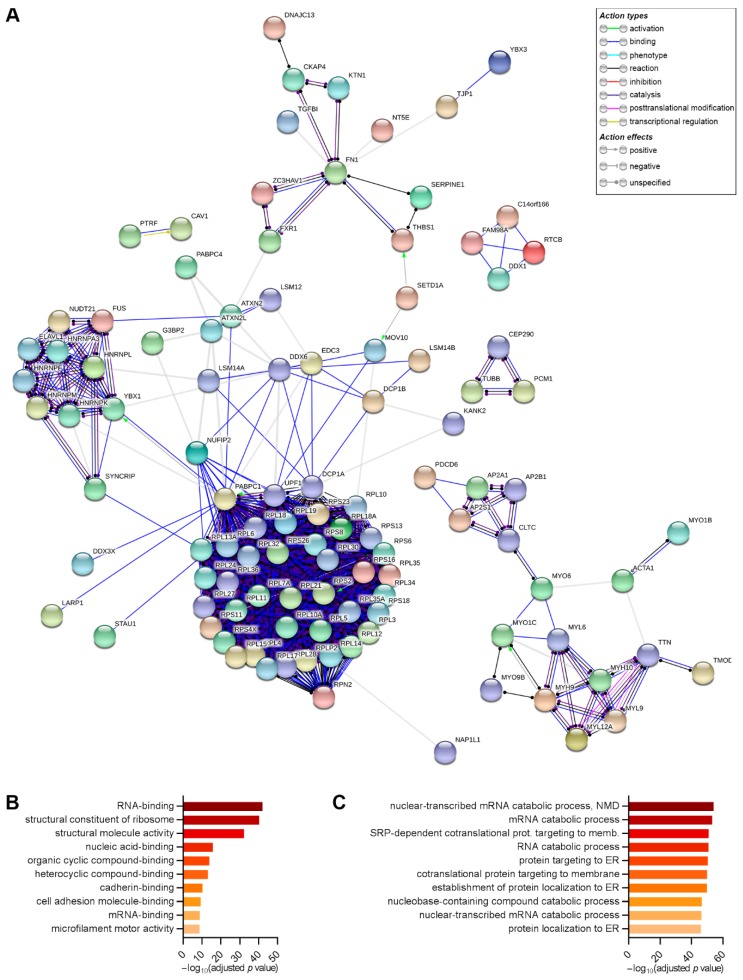
Interaction mapping and gene ontology of the proteins identified by coimmunoprecipitation with DDX6 in the hASCs that were not induced or induced to adipogenesis or osteogenesis for 24 h. (**A**) The interactions among the proteins identified by coimmunoprecipitation with DDX6 were analyzed using the STRING database. The map (produced using STRING, version 11.0) shows the proteins with predicted interactions of high confidence (STRING score > 0.7). (**B**) Enriched terms found by gene ontology analysis in the molecular function category (gProfiler). The bar graph represents the 10 terms with the lowest adjusted *p* value. (**C**) Enriched terms found by gene ontology analysis in the biological processes category (gProfiler). The bar graph represents the 10 terms with the lowest adjusted *p* value. ER = endoplasmic reticulum and NMD = nonsense-mediated decay.

**Figure 7 ijms-21-02607-f007:**
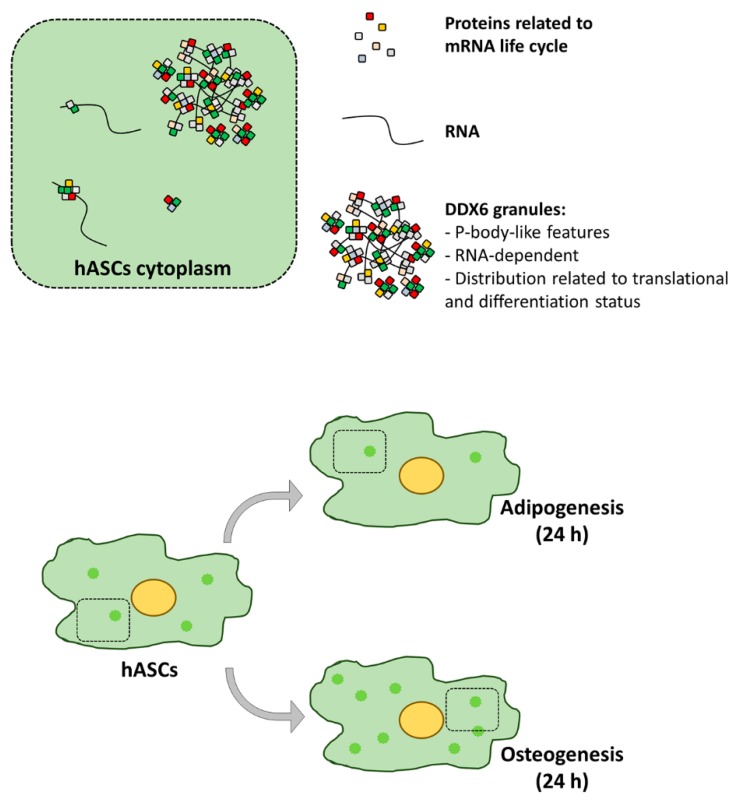
DDX6 complex composition and distribution in the hASCs that were not induced and those induced to adipogenesis or osteogenesis for 24 h.

**Table 1 ijms-21-02607-t001:** Sex, age, weight, height and body mass index (BMI) of the donors of adipose tissue used in this study.

Donor	Sex	Age (years)	Weight (kg)	Height (m)	BMI (kg/m^2^)	Analysis
A	Female	46	61	1.59	24.1	IP, IFF
B	Female	17	64	1.58	25.6	IP, IFF
C	Female	46	57	1.58	22.8	IP, IFF

IP = Immunoprecipitation; IFF = immunofluorescence.

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
