# Peer review of "DDX6 Helicase Behavior and Protein Partners in Human Adipose Tissue-Derived Stem Cells during Early Adipogenesis and Osteogenesis"

_ijms, 2020, doi:10.3390/ijms21072607_

Round 1

Reviewer 1 Report

This is interesting, actual and well done work, I have only a minor comment regarding hASC characteristics (Fig. S1 B), i.e. "Cells show hydroxyapatite deposits" - that's supposition as you do not show any data indicating these are specific HA deposits - please verify. 

Also, please provide some extra details regarding the differentiation treatments - at which passages cells were analyzed, when exactly during culture differentiation media were added (a day after cell seeding or upon cell seeding?) - this may be important to take into account cell status

Author Response

Dear reviewer, thank you for taking time to read and analyze our work. Your comments were very helpful. We carefully revised our manuscript to elucidate the raised questions. Please find herein the answers to the minor points.

POINT 1: This is interesting, actual and well done work, I have only a minor comment regarding hASC characteristics (Fig. S1 B), i.e. "Cells show hydroxyapatite deposits" - that's supposition as you do not show any data indicating these are specific HA deposits - please verify.

Response: To confirm bone mineralization, we used OsteoImage Mineralization kit, which, according to the manufacturer, specifically binds to the hydroxyapatite portion of bone-like nodules deposited by cells. In order to better explain this in the supplementary figure 1, we changed the legend of Figure S1.

Figure S1. Results from the analysis of the differentiation potential of the hASCs. (A) Microscopy analysis of the hASCs induced to adipogenesis for 28 days. After treatment, lipid droplets, which are characteristic of adipocytes, can be visualized by phase contrast and are stained green by Nile red. Nuclei were stained with DAPI. (B) Results from the microscopy analysis of the hASCs induced to osteogenesis for 21 days. After treatment, OsteoImage Mineralization Assay was used to detect bone-like nodules in the cell cultures. The kit consists of a fluorescent green staining reagent that specifically binds to the hydroxyapatite portion of these nodules. Nuclei were stained with DAPI.

Moreover, we included further explanation in the Materials and Methods section:

Materials and Methods; Subjects and cell culture; 4th paragraph: […] The efficiency of adipogenic differentiation was determined by assessing the cytoplasmic accumulation of triglycerides with Nile red staining. The osteogenic differentiation was assessed using the OsteoImage™ Mineralization Assay (Lonza), which contains a reagent that specifically binds and stains in fluorescent green the hydroxyapatite portion of the bone-like nodules deposited by the cells.

We are also sending in attachment the technical datasheet of the product.

POINT 2: Also, please provide some extra details regarding the differentiation treatments - at which passages cells were analyzed, when exactly during culture differentiation media were added (a day after cell seeding or upon cell seeding?) - this may be important to take into account cell status

Response: Dear reviewer, thank you for your observation. The differentiation treatment was performed with hASCs in passage 4-5 and was initiated when cells reached 80% confluence. The adipogenic differentiation was induced using hMSC Adipogenic Differentiation Medium for 3 days and hMSC Adipogenic Maintenance Medium for 4 days (hMSC Adipogenic Bullet Kit, Lonza); and this cycle was repeated for a total of 28 days. For osteogenic differentiation, the cells were treated with Osteogenic Differentiation BulletKit (Lonza) medium for 21 days.

We included further explanation concerning the differentiation methodology in the main text.

Materials and Methods; Subjects and cell culture; 4th paragraph: […] To assess the adipogenic and osteogenic differentiation potential of the isolated hASCs, we used hASCs in passages 4-5. The differentiation treatment was initiated when the cells reached 80% confluence. The adipogenic differentiation was induced using hMSC Adipogenic Differentiation Medium for 3 days and hMSC Adipogenic Maintenance Medium for 4 days (hMSC Adipogenic Bullet Kit, Lonza); and this cycle was repeated for a total of 28 days. For osteogenic differentiation, the cells were treated with Osteogenic Differentiation BulletKit (Lonza) medium for 21 days.

Reviewer 2 Report

This is a very well described study on characterising the protein complexes associated with DDX6 in human adipose tissue-derived stem cells (hASCs) and analysing the dynamics of this helicase under different conditions.

The rationale of the study is clearly supported by the introduction. The materials & methods are usually adequately described, the results are clear and easy to follow. The discussion is well written and helps the reader comprehend the new findings presented in this work.

I only have a few minor comments.

1) Page 9, line 252, western blot should not start with a capital W.

2) Page 17 - Could the authors please elaborate how exactly cell lysate was prepared after adding denaturing puffer? (i.e. mechanical trituration, sonication, etc).

3) Page 18 - How exactly counting the granules was performed? (i.e. manually or software-based)

Author Response

Dear reviewer, thank you for taking time to read and help improving our work. Your comments were helpful. We carefully revised and edited our manuscript to answer to your questions. Please find herein the answers to the minor points.

POINT 1: Page 9, line 252, western blot should not start with a capital W.

Response: As requested, we corrected the sentences:

Figure 4 legend: […] Quantification of DDX6 protein expression (by western blotting) in hASCs maintained […]

Results; topic 2.3; 3rd paragraph: […] Then, using western blot analysis, we verified […]

Results; topic 2.5; 1st paragraph: […] performed western blotting to identify the presence […]

POINT 2: Page 17 - Could the authors please elaborate how exactly cell lysate was prepared after adding denaturing puffer? (i.e. mechanical trituration, sonication, etc).

Response:  The buffer solution used to prepare the samples for western blot analysis contains: 160 mM Tris-HCl, pH 6.8; 4% SDS; 10% b-mercaptoethanol; 24% glycerol; and 0.02% bromophenol blue; which promotes both cell lysis and denatures proteins. Moreover, after adding the lysis/denaturing buffer, the cells were scraped, the cell extract was collected and incubated at 95o C for 10 minutes to complete cell lysis and protein denaturation. To clarify this process to the readers, we corrected the methodology text:

 Materials and Methods; topic 4.2; 1st paragraph: […] The cells were washed with PBS and scraped after the addition of a buffer solution (160 mM Tris-HCl, pH 6.8; 4% SDS; 10% b-mercaptoethanol; 24% glycerol; and 0.02% bromophenol blue) that promotes cell lysis and denatures proteins. The cell extract was collected, incubated at 95°C for 10 min and subjected to SDS-PAGE. Then, the proteins were transferred to nitrocellulose membranes.  […]

POINT 3: Page 18 - How exactly counting the granules was performed? (i.e. manually or software-based)

Response: Thank you for your observation. The counting of the granules was performed manually. First, 20 images of cells in each condition were captured. Using LAS AF software, a cutoff was established based on light intensity (using the threshold tool) such that only the granules (brighter structures) were exhibited. The same threshold was applied for all the conditions, and the number of granules in each cell was counted manually.

We added this observation in the main text:

Materials and Methods; topic 4.3; 2nd paragraph: […] and the number of granules in each cell was counted manually. […]

Reviewer 3 Report

In the manuscript by Marcon et al., the authors characterized the localization of DDX6 helicase in human adipose tissue derived stem cells (hASCs). Using immunofluorescence, they show that DDX6 is mainly localized in the cell cytoplasm either dispersed or concentrated in granules. The DDX6 granules disassembled upon RNase treatment suggesting their maintenance using an RNA dependent mechanism. The localization of DDX6 granules changed under oxidative stress conditions and colocalized with TIA1 and PABP, which are proteins known to colocalize with stress granules (SGs). The DDX6 granules also partially colocalized with DCP1A, a protein typically found in P-bodies, both with and without stress induction suggesting that the DDX6 granules could exhibit P-body like behavior.

To investigate this further, they used OPP to reduce the translational activity and mRNA release from polysomes and saw that the number of DDX6 granules per cell increased upon OPP treatment suggesting that the DDX6 granules assembled due to polysome disruption. These granules neither colocalized with a known characteristic SG protein, TIAR, which was found localized mostly in the nucleus, nor were they enriched with DRIPs known to accumulate in SGs after polysome disassembly. However, upon induction of stress using sodium arsenite treatment, the DDX6 granules colocalized with SGs and accumulated DRIPs consistent with the characteristic of P-bodies.

The authors further looked at the distribution of DDX6 granules in hASCs upon differentiation into adipogenic and osteogenic lineage. While the number of granules per cell decreased upon adipogenesis, they increased upon osteogenesis. This led them to investigate the abundance levels of DDX6 mRNA and protein in undifferentiated and differentiated state into osteogenesis and adipogenesis. From their previous RNA seq analysis on hASCs, they found no difference in total or polysome associated DDX6 mRNA levels in either of the conditions. They verified the RNA seq results using western blot and observed no changes in levels of protein as well, suggesting that the difference in DDX6 granule deposition during adipogenic and osteogenic differentiation was not due to regulation of DDX6 protein levels.

The DDX6 granules accumulated during differentiation also exhibited similar P-body like characteristics as observed earlier. The granules disassembled upon RNAse treatment and did not colocalize with DRIPs after OPP treatment. The authors analyzed the distribution of proteins associated with P-bodies and SG in hASCs induced to become adipocytes and osteocytes. TIA1 and PABP showed no difference in the localization pattern upon differentiation. On the other hand, DCP1A formed granules 24 hours post differentiation into either adipogenic or osteogenic lineage. However, not all DDX6 granules colocalized with DCP1A granules suggesting a characteristic like that of P-bodies.

Finally, the authors investigated the components of the complexes associated with DDX6 using immunoprecipitation and mass spectrometry on hASCs in undifferentiated and differentiated state. They were able to identify many of the proteins previously known to be associated with DDX6 in different cell types, in all the conditions and identified 66 out of 142 novel proteins that were not known previously. They identified proteins that were differentially associated with DDX6 in different conditions such as ELAV1 which was more enriched in non-induced samples compared to adipogenic cells and RPN2 which was more enriched in osteogenic cells compared to non-induced cells. GO analysis on these associated proteins revealed that most of the proteins associated with DDX6 were involved in RNA binding, RNA life cycles and RNA catabolic processes.

To summarize, the manuscript sheds light on the dynamic association of DDX6 with P-bodies in human adipose tissue-derived stem cells during adipogenesis and osteogenesis. This is a very interesting area of study. The complex role of P-bodies and the role of DDX6 in particular is complex and relatively little is understood. However, this study provides little insight into how DDX6 is important for the differentiation process and maintenance/functionality of osteocytes and adipocytes.

Major issues-

  1. The manuscript lacks any evidence of homogeneity amongst the hASC and efficiency of differentiation into osteocytes and adipocytes for each experiment. The fact that the adipocyte samples in figure 5 clustered first with NI samples and sometimes with osteocyte samples in different cases suggests inefficient and non-reproducible differentiation.
  2. The authors fail to show if the trend in P-body-like behavior of DDX6 upon induction of differentiation is specific to fully differentiated osteocytes and adipocytes or if the undifferentiated cells exhibited a similar pattern. How are you defining differentiated cells in your assay?
  3. The second half of the paper focuses on the dynamics of DDX6 granule formation in undifferentiated and differentiated states but fails to explore any functional role in adipogenesis and osteogenesis.
  4. The authors do not validate using alternative methods for the mass spectrometry results, particularly for the 66/142 novel proteins that they have identified.
  5. The authors have not demonstrated a role for their novel proteins to see which, if any, are required for granule formation and the differentiation process.
  6. The section where the authors talk about DDX6 fractionating with ribosomes upon differentiation fails to contribute any significant insight to the manuscript and lacks further exploration. This piece of evidence hints towards an active role of DDX6 in promoting translation during differentiation. It would be interesting to know if the transcripts bound to the DDX6-ribosome complexes correspond to genes important for the differentiation process.
  7. Figures 4c and 4d seem redundant. Can these results be combined to convey that the number of DDX6 granules change upon differentiation? How are they different?
  8. The protein interactome in Figure 6a does a poor job at highlighting any interesting findings that could contribute to any further investigations. Is there a way to elaborate on these results?

Minor issues- 

The manuscript has some typos and errors. Below are a few examples. Line #’s listed with corrections in parentheses.

  1. 69 DiStefano and collaborators demonstrated that P-body homeostasis was important "for" the balance
  2. 72 which is a component of P-bodies but is not essential "for" its maintenance, did not affect cell
  3. 128 Figure 2. Localization of DDX6 and SG or P-body proteins in the hASCs "is" not induced by stress and
  4. 417 3 hours. Here, we used sodium arsenite for 30 min, which leads to oxidative stress. Further "studies" would
  5. 423 induced “to” adipogenesis or osteogenesis for 24 hours.
  6. 463 immunofluorescence in a previous work, "which" might suggest that this protein may connect these

Author Response

Dear reviewer, thank you for taking time to read and analyze our work. Your comments were helpful and will help to improve our work. We carefully revised and edited our manuscript according to your recommendations. Please find herein the answers to the major and minor points.

Major issues-

POINT 1: The manuscript lacks any evidence of homogeneity amongst the hASC and efficiency of differentiation into osteocytes and adipocytes for each experiment. The fact that the adipocyte samples in figure 5 clustered first with NI samples and sometimes with osteocyte samples in different cases suggests inefficient and non-reproducible differentiation.

Response: Thank you for your observation. The process of adipogenic and osteogenic differentiation of hASCs using the induction medium from Lonza takes over 20 days to be accomplished, as described in the manufacturer datasheet and in the item 4.1 of the Materials and Methods section. As the aim of our work was to evaluate the early response of hASCs to the induction treatment, we analyzed the dynamics and behavior of DDX6 after the first 24 hours of treatment. We had previously demonstrated that the hASCs induced to adipogenesis or osteogenesis for 24 hours have an important change in gene expression regulation both at transcriptional and translational levels, but are not committed with the differentiation into osteogenic or adipogenic lineages (Robert et al., 2018; Spangenberg et al., 2013). The same observation was also demonstrated by at least other group (Scheideler et al., 2008). Then the use of specific markers to evaluate the homogeneity and efficiency of the differentiation process in at such an early stage of the differentiation process would be extremely difficult. Therefore, we chose to analyze the overall early response of the hASCs population to the induction to adipogenesis or to osteogenesis.

Moreover, hASCs are known to be composed by a non-homogeneous population (Baer and Geiger, 2012; Şovrea et al., 2019), that may likewise have a non-homogeneous response to the induction stimulus. Nevertheless, this heterogeneous population of hASCs is a putative source for use in regenerative medicine (Kocan et al., 2017; Şovrea et al., 2019). Then studies focusing on the population of primary hASCs cultures (as our study) are interesting to understand the behavior and the response of this population in situations as stress or differentiation induction, for example.

We believe that this is an important discussion to be presented. Then we included a new paragraph in our manuscript:

Discussion; 5th paragraph: […] The immunofluorescence analysis performed focused in the overall behavior of the hASCs population, which is characterized as composed by a non-homogeneous population [40,41]. As previously demonstrated, 24 hours of induction treatment is not enough to compromise the hASCs specifically with the adipogenic [42,43] or the osteogenic [30,43] differentiation. Hence the use of specific markers to identify which cells have already triggered the differentiation process in this scenario would be difficult. In the future, it would be interesting to perform time lapse experiments to follow DDX6 localization along the differentiation course to evaluate a closer relationship between this process change in DDX6 distribution.  […]

Regarding the clustering analysis, we also analyzed the overall protein composition of the complexes associated with DDX6, independent of being in granules or dispersed in the cytoplasm. Note that we are careful in referring to this as complexes associated with DDX6 or DDX6-containing complexes. Again, we did not distinguish the complexes found dispersed in the cell cytoplasm or in granules. The aim of this analysis was to characterize the proteins associated with DDX6 in stemness stage and after 24 hours of induction to adipogenesis or to osteogenesis. And the results obtained suggest that the overall protein composition of these complexes is very similar in the three conditions analyzed. Nevertheless, it is possible that these proteins may be differentially distributed in specific complexes. We believe that it is valuable to explicit this issue in our discussion; thus, we have included a line stating this idea.

Discussion; 7th paragraph: […] The results obtained suggested that the complexes associated to DDX6 in hASCs have a similar protein content in the conditions analyzed. Nevertheless, it is possible that these proteins may be differentially distributed in specific complexes. To analyze these specific complexes, it would be necessary, for example, to separate fractions of complexes or to perform the immunoprecipitation of other proteins to characterize their complexes and to compare with DDX6 complexes. […]

POINT 2: The authors fail to show if the trend in P-body-like behavior of DDX6 upon induction of differentiation is specific to fully differentiated osteocytes and adipocytes or if the undifferentiated cells exhibited a similar pattern. How are you defining differentiated cells in your assay?

Response: As discussed previously, we focused in the early steps of the adipogenic and the osteogenic differentiation of hASCs. In this stage (24 h after induction), there are no known ways or specific markers to identify which cells already entered the differentiation process and which ones did not. The idea was to study the very early redistribution and possible differential association of DDX6 to proteins in the different situation analyzed. As also proposed in answer to point 1, future studies focusing in the analysis of DDX6 distribution over the differentiation time course would be interesting to verify a possible correlation between these processes and maybe stablish DDX6 distribution as a marker for adipogenic or osteogenic differentiation on hASCs.

POINT 3: The second half of the paper focuses on the dynamics of DDX6 granule formation in undifferentiated and differentiated states but fails to explore any functional role in adipogenesis and osteogenesis.

Response: Thank you for your comment. The aim of our work was to characterize DDX6 expression, distribution and behavior in hASCs. Even though we did not explore any functional role of DDX6 complexes in adipogenesis and osteogenesis, we believe that our work may function as a dataset for other groups to explore these features. Based on the list of proteins that we described as associated with DDX6 in hASCs, the scientific community have a very precious set of data that might be the focus of other groups in order to study the roles played by specific proteins.

POINT 4: The authors do not validate using alternative methods for the mass spectrometry results, particularly for the 66/142 novel proteins that they have identified.

Response: To identify the proteins associated with DDX6 complexes, we chose first to stablish a quite stringent protocol for immunoprecipitation. Then, we performed several controls. We analyzed by mass spectrometry the protein content of several fractions obtained during immunoprecipitation of DDX6: the input (cell extract), the flow-through (cell extract after the immunoprecipitation process), and the precleaning beads, and we also used the same protocol with an antibody for a nonrelated protein (anti-histone H3). In the input and flow-through analysis, the signal obtained for DDX6 corresponded to less than 0.05% of the signal obtained in the immunoprecipitation sample, showing that the protocol used was efficient for the enrichment of this protein. And DDX6 was not found to be associated with the precleaning beads and was not found with immunoprecipitated histone H3 (Table S2), showing that the methodology was also specific.

Next, we performed the DDX6 immunoprecipitation in hASCs maintained in noninductive medium or induced to adipogenesis or osteogenesis for 24 hours. The analysis of the mass spectrometry data was also performed using stringent criteria to reduce false positives: Proteins identified only by site, reverse, and potential contaminants were removed. The proteins identified in the analysis of the precleaning beads were also removed from the coimmunoprecipitation list because of their potential for unspecific binding. We considered as identified protein in a sample when having at least 2 unique peptides in at least one of the replicates.

We then used the STRING database to analyze the predicted interaction among the proteins identified. This analysis confirmed that most of the proteins that immunoprecipitated with DDX6 had interactions predicted for binding, reinforcing the accuracy of the coimmunoprecipitation results.

To further confirm the mass spectrometry data, we now performed western blot analysis and confirmed the presence of DDX6, Dcp1a and RPL30 (a previously unknown partner of DDX6) in the immunoprecipitation sample. We added in the manuscript the following information:

Results; subitem 2.6; 2nd paragraph: […] To further confirm the results obtained by mass spectrometry analysis, we performed western blot analysis of the immunoprecipitation of DDX6 sample and confirmed the presence of DDX6, Dcp1a and RPL30 (Figure S8 A, B and C).

Materials and Methods; subitem 4.2; 2nd paragraph: […] The nitrocellulose membrane was blocked (1 hour incubation with 5% milk in Tris buffered saline - TBS), washed and incubated with the primary antibody (anti-DDX6, MBL, PD009, 1:2500; anti-DCP1A, Santa Cruz, sc-100706, 1:200; anti-RPL30, Abcam, ab170930). After incubation with the suitable secondary antibody (anti-mouse IgG-alkaline phosphatase, Sigma, 1:1000; anti-rabbit IgG-peroxidase, produced in goat, Sigma, 1:2500), the membranes were analyzed with AP buffer, BCIP e NBT or with a Novex® ECL HRP chemiluminescent kit. The signal intensity was quantified with ImageJ software.

Supplementary material: Figure S8. Western blot analysis of the proteins found in DDX6 immunoprecipitation. Western blot analysis using the protein extract from DDX6 immunoprecipitation from hASCs kept in noninductive medium and submitted to labeling with (A) anti-DDX6, (B) anti-Dcp1a and (C) anti-RPL30 antibodies. (D) Western blot analysis using the protein extract from DDX6 immunoprecipitation from hASCs maintained in noninductive medium (1) or induced to adipogenesis (2) or osteogenesis (3) for 24 hours and submitted to labeling with anti-DDX6. Molecular weight is presented in kDa.

To further confirm the data obtained, we also confirmed by western blot the presence of the ribosomal protein RPL30 in the DDX6 immunoprecipitation samples from hASCs noninduced and induced to adipogenesis or to osteogenesis. We added in the manuscript the following information:

Results; subitem 2.6; 6th paragraph: […] We further confirmed by western blot the presence of RPL30, a protein from the large ribosomal subunit in the extracts obtained from the immunoprecipitation of DDX6 in the three conditions analyzed: noninduced, induced to adipogenesis and induced to osteogenesis (Figure S8 D). Notably, RPL30 had not been previously identified in the studies of Ayache (HEK293T cells, treated with RNase inhibitor), DiStefano (hiPSCs) and Wang (epidermal progenitors).

POINT 5: The authors have not demonstrated a role for their novel proteins to see which, if any, are required for granule formation and the differentiation process.

Response: We agree that a deeper investigation focusing on novel functions for the proteins found un DDX6 granules would be very interesting. Nevertheless, this approach would demand several experiments and analysis that were not the goal of our work at this time. Our aim was to make a primary characterization of DDX6 behavior and to describe the protein content of the complexes associated with this helicase in hASCs. We believe that the information provided in our manuscript may be helpful as a database for studies focusing both in DDX6 function and/or in hASCs biology.  We are conscious that our work is descriptive for the most part, but we are undoubtedly leaving open doors to mechanistic studies in the future.

POINT 6: The section where the authors talk about DDX6 fractionating with ribosomes upon differentiation fails to contribute any significant insight to the manuscript and lacks further exploration. This piece of evidence hints towards an active role of DDX6 in promoting translation during differentiation. It would be interesting to know if the transcripts bound to the DDX6-ribosome complexes correspond to genes important for the differentiation process.

Response: Regarding the polysome analysis, we agree that the finding of DDX6 fractionating with polysomes may suggest a possible role in translation. But we can not exclude the possibility that DDX6 may be associated to a complex that comigrates with polysomes in the sucrose gradient but is not directly associated with the polysomes.

We also agree that the identification of the transcripts associated with DDX6-ribosome complexes in hASCs would be very interesting. In parallel with the proteomic analysis, we also developed the isolation of the RNA fraction associated with DDX6 complexes. But the protocols, experiments (RNA-Seq) and analysis of this data are very complex. Moreover, this study requires several quality check steps and additional functional experiments to confirm the results obtained, which were not the aim of this manuscript.

POINT 7: Figures 4c and 4d seem redundant. Can these results be combined to convey that the number of DDX6 granules change upon differentiation? How are they different?

Response: Thank you for your observation. We agree that figures 4C and 4D are similar, but we do not believe that they are redundant. Our aim was to analyze the dynamic of DDX6 in hASCs noninduced, induced to adipogenesis or to osteogenesis. To make sure that the phenomenon observed was due to cell treatment and not donor specific, we analyzed cells from three different donors. And, for each donor, we analyzed 20 images for each condition. The boxplot of the values found for each condition of each donor is represent in the figure S3. It was notable that the values found did not have a symmetric distribution. Nevertheless, by statistical analysis, we verified that, for each donor, the number of granules per cell were influenced by differentiation induction. To analyze and compare the data obtained from the three donors, we calculated the mean and the median number of granules per cells for each condition of each donor and performed the statistical analysis. The mean number considers the entire population and was important to investigate the overall response of this heterogeneous population to the different treatments. On the other hand, the median represents the central number and is not as influenced by outliers. Then, we also compared the median number of granules per cell found for each donor in each condition. Then, each analysis shows different approaches concerning the quantification of DDX6 granules per cell. And, withal, the three analysis reinforce the conclusion that the number of DDX6 granules per cell is influenced by the induction to adipogenesis or to osteogenesis. 

To further clarify this observation, we edited the manuscript:

Results; subitem 2.3; 2nd paragraph: […] In the hASCs induced to adipogenesis, the percentage of cells that contained DDX6 granules (Figure 4 B) and the mean number of granules per cell were both reduced (Figure 4 C and D). On the other hand, when osteogenesis was induced, the mean number of DDX6 granules per cell (Figure 4 C and D) was increased. In addition, the number of granules per cell was variable in all culture conditions (not induced, induced to adipogenesis and induced to osteogenesis), showing an uneven distribution (Figure S3 A), and the mean number of granules per cell was lower higher than the median number of granules per cell (Figure 4 C and D). Nevertheless, all the analysis performed demonstrated that the early steps of adipogenic or osteogenic induction involved a change in the distribution of DDX6 granules.  […]

POINT 8: The protein interactome in Figure 6a does a poor job at highlighting any interesting findings that could contribute to any further investigations. Is there a way to elaborate on these results?

Response: As described in our response to point 4, we used the STRING database to explore which interactions were predicted among the proteins identified in DDX6 complexes. The results obtained demonstrated that most of the proteins identified had predicted interactions for binding, catalysis and reaction, with a calculated PPI (protein-protein interaction) enrichment p-value < 1.0e-16. As part of our work focused on the description of the protein content of DDX6 complexes by immunoprecipitation, we believe that this information is important to confirm the accuracy of the coimmunoprecipitation results.

Minor issues-

The manuscript has some typos and errors. Below are a few examples. Line #’s listed with corrections in parentheses.

Dear reviewer, thank you for your observations. We had submitted our manuscript to edition to proper English language, grammar, punctuation, spelling and overall style by American Journal Experts. Nevertheless, it is possible that some typos and errors may persist. We included your suggestions as follows:

69 DiStefano and collaborators demonstrated that P-body homeostasis was important "for" the balance

Response: We corrected the text: […] DiStefano and collaborators demonstrated that P-body homeostasis was important for the balance […]

72 which is a component of P-bodies but is not essential "for" its maintenance, did not affect cell

Response: We corrected the text: […] which is a component of P-bodies but is not essential for its maintenance, did not affect cell differentiation […]

128 Figure 2. Localization of DDX6 and SG or P-body proteins in the hASCs "is" not induced by stress and

Response: Thank you for your observation. In this specific sentence, we stand for our previous sentence: […] Figure2. Localization of DDX6 and SG or P-body proteins in the hASCs not induced by stress and after stress induction[…]. Because, in this case, this is the title of the figure and consists in a description of the content of the image, not the result.

417 3 hours. Here, we used sodium arsenite for 30 min, which leads to oxidative stress. Further "studies" would

Response: We corrected the text: […] Further studies would allow to determinate whether […]

423 induced “to” adipogenesis or osteogenesis for 24 hours.

Response: We corrected the text: […] those induced to adipogenesis or osteogenesis for 24 hours […]

463 immunofluorescence in a previous work, "which" might suggest that this protein may connect these

Response: We corrected the text: […] immunofluorescence in a previous work, which might suggest that this protein may connect […]

REFERENCES

Baer, P.C., Geiger, H., 2012. Adipose-Derived Mesenchymal Stromal/Stem Cells: Tissue Localization, Characterization, and Heterogeneity. Stem Cells Int. 2012, 1–11. doi:10.1155/2012/812693

Kocan, B., Maziarz, A., Tabarkiewicz, J., Ochiya, T., Banaś-Ząbczyk, A., 2017. Trophic Activity and Phenotype of Adipose Tissue-Derived Mesenchymal Stem Cells as a Background of Their Regenerative Potential. Stem Cells Int. 2017, 1653254. doi:10.1155/2017/1653254

Robert, A.W., Angulski, A.B.B., Spangenberg, L., Shigunov, P., Pereira, I.T., Bettes, P.S.L., Naya, H., Correa, A., Dallagiovanna, B., Stimamiglio, M.A., 2018. Gene expression analysis of human adipose tissue-derived stem cells during the initial steps of in vitro osteogenesis. Sci. Rep. 8, 4739. doi:10.1038/s41598-018-22991-6

Scheideler, M., Elabd, C., Zaragosi, L.E., Chiellini, C., Hackl, H., Sanchez-Cabo, F., Yadav, S., Duszka, K., Friedl, G., Papak, C., Prokesch, A., Windhager, R., Ailhaud, G., Dani, C., Amri, E.Z., Trajanoski, Z., 2008. Comparative transcriptomics of human multipotent stem cells during adipogenesis and osteoblastogenesis. BMC Genomics 9, 340. doi:10.1186/1471-2164-9-340

Şovrea, A.S., Boşca, A.B., Constantin, A.M., Dronca, E., Ilea, A., 2019. State of the art in human adipose stem cells and their role in therapy. Rom. J. Morphol. Embryol. 60, 7–31.

Spangenberg, L., Shigunov, P., Abud, A.P.R., Cofré, A.R., Stimamiglio, M.A., Kuligovski, C., Zych, J., Schittini, A. V., Costa, A.D.T., Rebelatto, C.K., Brofman, P.R.S., Goldenberg, S., Correa, A., Naya, H., Dallagiovanna, B., 2013. Polysome profiling shows extensive posttranscriptional regulation during human adipocyte stem cell differentiation into adipocytes. Stem Cell Res. 11, 902–912. doi:10.1016/j.scr.2013.06.002

Round 2

Reviewer 3 Report

The authors were very responsive to our criticism and have reasonably addressed our concerns. Thank you.